# Structural Investigation of Magnesium Complexes Supported by a Thiopyridyl Scorpionate Ligand

**DOI:** 10.3390/molecules27144564

**Published:** 2022-07-18

**Authors:** Matthew P. Stevens, Emily Spray, Iñigo J. Vitorica-Yrezabal, Kuldip Singh, Vanessa M. Timmermann, Lia Sotorrios, Fabrizio Ortu

**Affiliations:** 1School of Chemistry, University of Leicester, University Road, Leicester LE1 7RH, UK; mps28@leicester.ac.uk (M.P.S.); emily.spray@hotmail.co.uk (E.S.); ks42@leicester.ac.uk (K.S.); vanessa.timmermann@siigroup.com (V.M.T.); 2Department of Chemistry, The University of Manchester, Oxford Road, Manchester M13 9PL, UK; inigo.vitorica@manchester.ac.uk; 3Institute of Chemical Sciences, Heriot-Watt University, Edinburgh EH14 4AS, UK

**Keywords:** organometallic chemistry, alkaline earth, inorganic, scorpionate, ligand design, DFT, buried volume calculations

## Abstract

Herein, we report the synthesis of a series of heteroleptic magnesium complexes stabilized with the scorpionate ligand tris(2-pyridylthio)methanide (Tptm). The compounds of the general formula [Mg(Tptm)(X)] (**1-X**; X = Cl, Br, I) were obtained via protonolysis reaction between the proligand and selected Grignard reagents. Attempts to isolate the potassium derivative K(Tptm) lead to decomposition of Tptm and formation of the alkene (C_5_H_4_N-S)_2_C=C(C_5_H_4_N-S)_2_, and this degradation was also modelled using DFT methods. Compound **1-I** was treated with K(CH_2_Ph), affording the degradation product [Mg(Bptm)_2_] (**2**; Bptm = {CH(S-C_5_NH_3_)_2_}^−^). We analyzed and quantified the steric properties of the Tptm ligand using the structural information of the compounds obtained in this study paired with buried volume calculations, also adding the structural data of HTptm and its CF_3_-substituted congener (HTptm^CF3^). These studies highlight the highly flexible nature of this ligand scaffold and its ability to stabilize various coordination motifs and geometries, which is a highly desirable feature in the design of novel organometallic reagents and catalysts.

## 1. Introduction

Bulky, multidentate ligands have found extensive use for the stabilization of alkaline earth (AE) metal complexes, owing to their ability to saturate the metal coordination sphere, preventing unwanted attacks from large donor groups [1,2,3,4]. Ligands of this type can be designed and tailored to offer one open face in the metal coordination sphere, which can be used for selective reactivity or further functionalization [5]. An example of the application of bulky, multidentate ligands in AE chemistry is the substituted tris(pyrazolyl)borate (Tp) architectures [6]; recently, the extremely bulky tris(3-*^t^*Bu-5-Me-pyrazole)borate (Tp*^t^*^Bu,Me^) ligand was used by Anwander and co-workers to stabilise the complex [Ca(Tp*^t^*^Bu,Me^)(Me)], which contains an unprecedented terminal Ca–Me bond. [7] This is a highly reactive functionality due to the large charge separation between the metal and carbon atom; in this case, the scorpionate Tp*^t^*^Bu,Me^ ligand is providing steric protection to the Ca–C bond [7], together with preventing dimerization and Schlenk equilibrium-type rearrangements [8].

Tp ligands are classic examples of scorpionate-type donors, which are capable of wrapping around the metal centre in a κ^3^-fashion. Scorpionate ligands can be further extended with the addition of a fourth coordination point which acts as an anchor. A good example of this approach is the tris[(1-isopropylbenzamidazol-2-yl)-dimethylsilyl]methyl ligand, (Tism^PriBenz^), which was utilized to form the terminal magnesium-methyl complex [Mg(Tism^PriBenz^)(Me)] and can be converted into the parent hydride [Mg(Tism^PriBenz^)(H)] [9]. The latter complex has found extensive applications as a catalyst for hydroboration and hydrosilylation of styrene and carbodiimides [10].

Tris(2-pyridylthio)methanide, {CH(S-C_5_NH_4_)_3_}^−^ (Tptm), is another example of a scorpionate ligand, which was first used by Kinoshita and co-workers to prepare various first-row transition metal complexes [11,12,13]. In further work, the same authors reported the preparation of heteroleptic Zn complexes of formula [Zn(Tptm)(X)] (X = Cl, Br); these species were obtained by reacting HTptm with ZnX_2_ (X = Cl, Br) in the presence of K_2_CO_3_ [14]. Parkin and co-workers subsequently reported a series of zinc complexes of general formula [Zn(Tptm)(X)] (X = Me, N(SiMe_3_)_2_), which were prepared via protonolysis from the corresponding homoleptic Zn bis-amide or bis-alkyl precursor; additionally, these functionalities can be exchanged to increase the reactivity of these complexes, by installing silyloxide or hydride functionalities [15]. An example of this is the reactivity of the heteroleptic zinc complex [Zn(Tptm){N(SiMe_3_)_2_}], which reacts readily with CO_2_ to give isocyanate complex [Zn(Tptm)(NCO)]; its hydride congener [Zn(Tptm)(H)] also reacts with CO_2_, affording formate species [Zn(Tptm)(O_2_CH)] [15].

Owing to the similarities between Zn^2+^ and Mg^2+^ for their coordination chemistry (Zn^2+^ 0.74 Å, Mg^2+^ 0.72 Å) [16], we were interested in extending the use of Tptm to Mg with the goal of isolating new heteroleptic complexes and exploring their reactivity towards small molecules. Ligand design is a crucial aspect in the preparation of reactive organometallic species, and the work of Kinoshita and Parkin with Tptm-supported complexes shows that this ligand system offers a certain degree of flexibility which could be exploited for the preparation of various heteroleptic species [14,15]. In the Zn complexes reported by these authors, [Zn(Tptm)(X)] (X = H, Me, N(SiMe_3_)_2_, F, Cl, Br, OH, OSiMe_3_, NCO), the Tptm ligand, switches between a *κ*^3^ and *κ*^4^ binding arrangement depending on the types of ligands bound to the metal centre [14,15,17,18]. We envisaged that a good starting point for our synthetic journey would be to prepare some model magnesium compounds and assess the coordination chemistry of the Tptm ligand. Whilst structural studies provide an excellent representation of coordination motifs, several other tools can be used to quantify the steric properties of the supporting ligands. The Tolman cone angle is probably the most famous parameter used for this purpose, and its scope and application have recently been revisited, as shown by Anwander’s cone-angle calculations carried out on scorpionate ligands bound to magnesium [19]. Other tools have also found applications in coordination and organometallic chemistry, such as the solid angle G parameter and buried volume (%V_bur_) [20,21,22]. The latter calculations have been used quite extensively over the last decade to quantify the amount of steric protection imparted by ligands, and they have entered routine use for ligand design [20,21]. Though initially developed for NHC ligands [23], these calculations have been applied extensively in Group 2 coordination chemistry, encompassing ligands with various binding modes, denticity, and steric features [24,25]. Therefore, we believed this tool could provide a quantitative assessment of the steric features of the Tptm ligand architecture in order to inform our ligand design strategy. Herein, we present the synthesis of a small family of heteroleptic magnesium halide complexes supported by Tptm, accompanied by a structural analysis based on the combination of single crystal XRD studies and %V_bur_ calculations.

## 2. Results

### Synthesis and NMR Characterization

The proligand tris(2-pyridylthio)methane (HTptm) was synthesized following minor modifications of literature methods reported by Parkin and co-workers (Figure 1) [15]. The preparation of HTptm was first reported by de Castro et al., who followed a slightly different methodology [26], though we could not reproduce their yields. It is also noteworthy that Kinoshita et al. used a different approach for the preparation of HTptm, which involved sonication of the reaction mixture and shorter reaction times. We were able to reproduce Parkin’s methodology very reliably and adapted their methods for synthesizing a more sterically demanding version of this architecture, comprising CF_3_ substituents in the 6-position on the pyridyl arms, i.e., HC[S{C_5_H_3_N(CF_3_)-6}]_3_ (HTptm^CF3^). However, its synthesis proved to be significantly more challenging than that of HTptm; our methodology affords HTptm^CF3^ in very poor yields (ca. 5%) therefore, it could only be made in small quantities sufficient for its full spectroscopic characterization.

The ^1^H NMR spectrum of HTptm^CF3^ displays a singlet at 8.11 corresponding to the CH methine proton (*c.f.* 7.90 ppm for HTptm), together with a triplet (7.68 ppm) and two doublets (7.40, 7.34 ppm) corresponding to protons H^b^, H^a^ and H^c^, respectively. In the ^13^C{^1^H} NMR spectrum of HTptm^CF3^, the quaternary CF_3_ signal appears as a single large quartet owing to the strong ^1^*J*_CF_ coupling (121.2 ppm, ^1^*J*_CF_ = 273 Hz), and coupling to ^19^F is also observed in the signal of quaternary carbon C^a^ (148.1 ppm, ^2^*J*_CF_ = 34.5 Hz). Additionally, a slight broadening (ν_½_ = 7.0 Hz) is observed for the C^b^-H signal (116.5 ppm), and the remaining C*^c^*-H and C*^d^*-H resonances are observed at 137.3 and 124.3 ppm, respectively. Finally, the ^19^F{^1^H} NMR spectrum of HTptm^CF3^ shows only one signal at 68.65 ppm, indicative of a single fluorine environment in solution.

Initially, we set out to prepare alkali metal salts of the Tptm ligand to be employed in salt elimination reactions. Parkin and co-workers reported a lithium derivative Li(Tptm) obtained from the reaction of HTptm with organolithium reagents [27]. We decided to attempt the preparation of the potassium analogue K(Tptm) from the reaction of K[N(SiMe_3_)_2_] with HTptm in toluene (Figure 2). DFT calculations suggest that this process, which also forms the free amine HN(SiMe_3_)_2_, is exergonic by 9.0 kcal/mol. However, the target compound decomposes readily to give the alkene (C_5_H_4_N-S)_2_C=C(C_5_H_4_N-S)_2_ and other by-products which could not be unequivocally identified. The detachment of a pyridyl arm from the Tptm scaffold has been previously observed by Kinoshita and co-workers when HTptm was reacted with FeI_2_ in the presence of Et_3_N and subsequent treatment with AgPF_6_, leading to an equilibrium between the carbene complex [Fe{C(S-C_5_H_4_N)_2_}(C_5_H_4_N-S)(I)] and [Fe(Tptm)(CH_3_CN)_2_][PF_6_] [28]. However, the formation of (C_5_H_4_N-S)_2_C=C(C_5_H_4_N-S)_2_ has not been previously reported. DFT calculations showed the formation of this decomposition product along with one equivalent of K(C_5_H_4_N-S) is favoured by 14.2 kcal/mol, with the decomposition reaction balanced assuming the formation of K(C_5_H_4_N-S).

We then decided to use a different approach for the synthesis of magnesium complexes involving a single-step protonolysis reaction. Protonolysis methodologies involving Grignard reagents are a common synthetic strategy in AE coordination and organometallic chemistry [29]. Hence, HTptm was reacted with selected Grignard reagents (i.e., ^i^PrMgCl, MeMgBr and MeMgI) in diethyl ether at room temperature (Figure 3), producing the target complexes [Mg(Tptm)(Cl)] (**1-Cl**), [Mg(Tptm)(Br)] (**1-Br**) and [Mg(Tptm)(I)] (**1-I**) with the concomitant formation of propane or methane. In all cases, the target compounds were obtained in excellent yields (>90%), and their formulations were confirmed via elemental analyses.

The ^1^H NMR spectra of **1-Cl**, **1-Br** and **1-I** bear many similarities and clearly show the three pyridyl protons in the aromatic region (Table 1). Only one set of signals is present in each case, thus hinting at the presence of a C_3_-symmetrical arrangement around the metal centre in solution for all complexes. Interestingly, there are some significant changes in the chemical shift of the pyridyl proton signals across the three compounds, though this behaviour does not appear to follow a specific trend (Table 1). For instance, proton H^a^ (Figure 1) resonates at 9.61 ppm for **1-Cl**, and the same signal is more shielded in **1-Br** (8.22 ppm) and more deshielded in **1-I** (9.88 ppm). However, the chemical shifts of the relative C-H signals in the ^13^C{^1^H} NMR spectra show very few variations across all these species. Nonetheless, the resonance of the methanide carbon is approximately the same in **1-Cl** and **1-Br** (15.0 and 15.1 ppm respectively), whilst it is significantly shifted in **1-I** (33.0 ppm); these signals are particularly difficult to detect and have not been reported before for analogous species such as [Zn(Tptm){N(SiMe_3_)_2_}] [15].

Halide functionalities can be replaced via simple salt elimination protocols with Group 1 reagents. With the idea of using this strategy to obtain a heteroleptic alkyl derivative of formula [Mg(Tptm)(CH_2_Ph)], **1-I** was reacted with one equivalent of K(CH_2_Ph) in THF at room temperature (Figure 3). Instead of isolating the target complex, a few crystals of [Mg(Bptm)_2_] (**2**; Bptm = {CH(S-C_5_NH_3_)_2_}^–^) were obtained together with an intractable mixture of products (Figure 4). The reasons for the formation of **2** are unclear; initially, the reaction produces a white precipitate which suggests that a salt elimination reaction is taking place, which generates KI. Therefore, we postulate that **2** is likely formed from the decomposition of the target complex [Mg(Tptm)(CH_2_Ph)], rather than from undesired reactivity of K(CH_2_Ph) with **1-I**. The ^1^HNMR spectrum of **2** displays one set of four proton signals in the aromatic region (δ_H_ = 5.98, 6.18, 6.52 and 8.32 ppm), indicative of the presence of a highly symmetric species in solution; accordingly, one single proton environment is observed for the hydrogen atom bound to the carbon donor on the two Bptm ligands (δ_H_ = 5.15 ppm). The ^13^C{^1^H} NMR spectrum also confirms the highly symmetrical nature of **2** in solution, showing a single environment for each aromatic carbon of Bptm (δ_C_ = 117.9, 121.6, 135.9, 147.2 and 166.5). Additionally, we were able to identify a signal resonating at 20.0 ppm, which can be assigned to the methanide carbon bound to Mg.

## 3. Structural Characterization and Buried Volume Calculations

XRD studies were carried out on single crystals of HTptm^CF3^ grown from an ethanol solution at room temperature. HTptm^CF3^ crystallizes in the *P*2_1_/*c* space group, and its molecular structure clearly shows that the pseudo-C_3_ symmetric conformation of HTptm is not possible with this species (Figure 2). Instead, one of the thiopyridyl arms sits below the plane of the sulfur atoms relative to the other two arms of the ligand. The H-C-S-C torsional angle of the thiopyridyl arm relative to the methanide C–H bond is 29.3(3)°, relative to 43.6° for HTptm, indicating that there is less steric clash about the central methanide in the substituted ligand relative to the parent, albeit with a considerable conformational difference.

Single crystals suitable for XRD studies were obtained for **1-Cl, 1-Br** and **1-I** (Figure 3), which confirmed the proposed connectivity in all cases. All three halide complexes crystallize as dimers, [{Mg(Tptm)(μ-X)}_2_], in which the halides bridge between two 6-coordinate Mg centres. In contrast to **1-Br** and **1-Cl**, the crystal structure of **1-I** also contains a monomeric molecule of **1-I** in addition to the dimer (Figure 4). This has a very high degree of disorder, and so whilst the connectivity is clear-cut, it is difficult to make any statistically valid comparisons or comments on bond distances and angles. However, it would appear that in the absence of a sixth coordination point, the magnesium adopts a distorted trigonal bipyramidal geometry. The Mg–C distances in **1-X** [Cl: 2.427(11); Br: 2.231(11); I: 2.231(10) Å] are similar to the analogous Mg–C bond in **2** [2.303(4) Å] and quite close to that observed for the related [Zn(Tptm)(Cl)] complex prepared by Kinoshita and co-workers [2.19(2), 2.213(6) Å] [14] and [Mg(Tptm){N(SiMe_3_)_2_}] [2.303(4) Å] [30]. The Mg–N_py_ bond distances [Cl: 2.186(11)–2.204(10); Br: 2.155(9)–2.219(9); I: 2.177(8)–2.182(8) Å] also fall within the range observed for analogous complexes [2.203(2)–2.305(2) Å] [30,31]. In all three **1-X** complexes, the geometry around the metal atom can be described as *mer-mer* with the three nitrogen atoms forming one meridional plane and the two halide atoms and methanide carbon forming the other.

In one instance, we were able to obtain the molecular structure of the THF adduct [Mg(Tptm)(I)(THF)] (**1-I∙THF**); **1-I∙THF** also adopts an octahedral geometry about magnesium, featuring the iodide ligand in an apical position in trans to the methanide carbon (Figure 5), analogously to monomeric **1-I**. The Mg–C distance [2.265(15) Å] is slightly longer than in unsolvated **1-I**, but still within the range of the values above. The Mg–N_py_ bond distance [2.213(13)–2.232(12) Å] is also slightly longer than the dimeric form but within the typical range [30,31].

Compound 2 crystallizes in the triclinic *P*–1 and features Mg in a 6-coordinate environment in what could be deemed as a distorted octahedral geometry (Figure 6). The complex is very symmetrical, with a perfect 180° angle between the two methanide carbon atoms, C(1) and C(1)^i^, and the metal centre. The four pyridyl nitrogen atoms complete the coordination sphere of magnesium by coordinating equatorially and with angles close to an ideal square planar arrangement [N–Mg–N 97.74(13)° and 87.26(13)°]. The Mg–C [2.235(5) Å] and Mg–N distances [2.201(4) Å and 2.226(3) Å] are consistent with those observed in **1-X**.

Working from crystal structures, the percentage buried volume, %V_bur_, was calculated using SambVca [20,21]. It was determined that the proligand HTptm, if coordinated to a metal without any change to the geometry, would occupy 66.7% of the coordination sphere of that metal (coordination axis, viewpoint and “top-down” steric maps are given in Figure 7, Figure 8 and Figure 9). In contrast, HTptm^CF3^ was found to have a significantly higher %V_bur_ of 75.7%; this is despite its arrangement in the solid state with one of the pyridyl arms flipped below the steric pocket. Buried volumes were also calculated for the Tptm ligand in the series of heteroleptic magnesium complexes **1-X** (Table 2). It was found the %V_bur_ of Tptm does not vary across the series of heteroleptic halide dimers (65.7%) and is close to the volume occupied by Tptm in the monomeric THF adduct **1-I·THF** (64.3%). However, %V_bur_ increases significantly in the case of monomeric **1-I** (75.3%), where the arrangement of Tptm forces the Mg centre in a 5-coordinate geometry (trigonal bipyramid) compared to a more open conformation in the 6-coordinate bridged halide dimers. For completeness, we also performed %V_bur_ calculations of Bptm by extracting coordinates from the molecular structure of **2** (Figure 10). As expected, Bptm displays a significantly smaller %V_bur_ (48.2%) compared to Tptm in any of the complexes reported in this work.

## 4. Discussion

Analysis of solid state structures combined with %V_bur_ calculations highlights the very flexible nature of the Tptm ligand. When Tptm binds a magnesium centre, its *κ*^4^ coordination mode affords two main conformations: ‘open’ and ‘closed’. In both conformations, the CS_3_ base of the tripodal ligand maintains a trigonal pyramidal geometry, whilst the pyridyl arms twist to form either an equatorial coordination arrangement around the Mg centre that is either *pseudo-*trigonal planar (‘closed’) or T-shaped (‘open’) (N∠N changes from 104.7(7)–132(2)° to 93.5(3)–96.1(3)°). As a result, complexes in the ‘open’ conformation feature a 6-coordinate magnesium centre, which is the coordination motif observed for **1-I·THF** and halide bridged dimers **1-X**. Conversely, in the ‘closed’ conformation (such as for monomeric **1-I** and [Mg(Tptm){N(SiMe_3_)_2_}] [24]), the CS_3_ base has a *pseudo* C_3_ symmetry that results in a better equatorial coordination saturation of the metal by the ligand pyridyl arms. Further flexibility still was observed in the analogous zinc complexes of general formula [Zn(Tptm)(X)] (X = H, Me, N(SiMe_3_)_2_) reported by Parkin and co-workers, wherein Tptm acts as a *k*^3^ donor, with one of the pyridyl pendant arms flipped below the steric pocket [15]. This is strongly reminiscent of the crystal structure of HTptm^CF3^. Therefore, we expect that Tptm^CF3^ will act exclusively as a *k*^3^ donor owing to its high steric demands.

## 5. Conclusions

In summary, we reported the synthesis and structural authentication of a series of heteroleptic magnesium-halide complexes (**1-X**; X = Cl, Br, I) supported by the scorpionate Tptm ligand, obtained via protonolysis reactivity between the proligand and selected Grignard reagents. Attempts to prepare the potassium derivative K(Tptm) led to facile decomposition of the ligand and identification of the alkene (C_5_H_4_N-S)_2_C=C(C_5_H_4_N-S)_2_; the energetic profile of this degradation pathway was also further analysed via DFT calculations which highlighted the exergonic nature of this process. Additionally, attempts to functionalise **1-I** with K(CH_2_Ph) also led to decomposition of Tptm and formation of **2**. Finally, we utilized the structural information obtained on all these complexes to assess the steric properties of Tptm and quantified these using buried volume calculations. Our studies highlight the high flexibility of the Tptm scaffold, which derives from the flexibility of its CS_3_ base and the ability of the pyridyl sidearms to twist into different conformations. As such, Tptm can accommodate different coordination environments and geometries, including different ancillary ligands. This flexibility is a highly desirable feature in the design of new reagents and catalysts, and this study will inform future ligand design to stabilise magnesium complexes with precise structure-function relationships.

## 6. Materials and Methods

### 6.1. General Methods

THF, Et_2_O, toluene and hexane were passed through columns containing molecular sieves, then stored over molecular sieves (THF) or over a potassium mirror (Et_2_O, hexane, toluene) and thoroughly degassed prior to use. Anhydrous benzene was purchased from Sigma Aldrich, stored over a potassium mirror and thoroughly degassed prior to use. For NMR spectroscopy, C_6_D_6_ and C_4_D_8_O were dried by refluxing over K, CDCl_3_ was dried by refluxing over CaH_2_; NMR solvents were then vacuum transferred and degassed by three freeze-pump-thaw cycles before use. NMR spectra were recorded on either a Bruker AVIII HD 400 or Bruker AVIII 500 spectrometer operating at 400.07/500.13 (^1^H), 100.60/125.78 (^13^C{^1^H}), 376.46 (^19^F{^1^H}) MHz. NMR spectra were recorded at 298 K unless otherwise stated and were referenced to residual solvent signals in the case of ^1^H and ^13^C{^1^H} experiments. FTIR spectra were recorded on a Bruker Alpha II spectrometer with a Platinum-ATR module. Elemental microanalyses were carried out by Elemental Microanalysis Ltd. Lawson’s reagent, 6-trifluoromethyl-2-pyridone, 2-mercaptopyridine, potassium hydroxide, bromoform, isopropylmagnesium chloride, methylmagnesium bromide, and methylmagnesium iodide were used as received. HTptm [25], 2-mercapto-6-trifluoromethylpyridine [32], and benzyl potassium [33] were prepared according to literature procedures. 

### 6.2. Synthesis

**HTptm^CF3^:** A solution of potassium hydroxide (2.313 g, 41.23 mmol) and 2-mercapto-6-trifluoromethylpyridine (4.751 g, 26.52 mmol) in ethanol (200 mL) was treated in a dropwise manner with bromoform (0.8 mL, 8.89 mmol) dissolved in ethanol (50 mL), then refluxed for 4 h. The solution was allowed to cool to room temperature and filtered. The solvent was removed in vacuo and the resultant residue was extracted into benzene, then passed through a silica plug. The solvent was removed in vacuo once more, and the resultant crude product recrystallized from ethanol. Crystals were isolated via decantation and washed with cold ethanol to give pure HTptm^CF3^ (0.250 g, 0.5 mmol, 5.2%) as large colourless crystals.

^1^H NMR (400 MHz, 298 K, CDCl_3_): δ_H_ (ppm) = 7.34 (3H, d, ^3^*J*_HH_ = 8.1 Hz, C_5_H_3_N-C*H*^c^), 7.40 (3H, d, ^3^*J*_HH_ = 7.6 Hz, C_5_H_3_N-C*H*^a^), 7.68 (3H, t, ^3^*J*_HH_ = 7.9 Hz, C_5_H_3_N-C*H*^b^), 8.11 (1H, s, S_3_C*H*^d^) ppm. ^13^C{^1^H} NMR (100 MHz, 298 K, CDCl_3_): δ_C_ (ppm) = 48.9 (s, S_3_*C*^f^H), 116.6 (br s, fwhm = 7.0 Hz, *C*_5_H_3_N-*C*^b^H), 121.2 (q, ^1^*J*_CF_ = 274 Hz, *C*^g^F_3_), 124.3 (*C*_5_H_3_N-*C*^d^H), 137.3 (*C*_5_H_3_N-*C*^c^H), 148.1 (q, ^2^*J*_CF_ = 35.8 Hz, *C*_5_H_3_N-*C*^a^CF_3_), 158.7 (*C*_5_H_3_N-*C*^f^S). ^19^F{^1^H} NMR (376 MHz, 298 K CDCl_3_): δ_F_ (ppm) = −68.65 (9F, s, C*F*_3_). Anal. calcd. for C_19_H_10_F_9_N_3_S_3_: C, 41.68%; H, 1.84%; N, 7.68%. Found: C, 41.63%; H, 1.79%; N, 7.64%. FTIR: $\tilde{\nu }$ (cm^−1^) = 2908, 1589, 1564, 1440, 1414, 1335, 1255, 1130, 1103, 988, 825, 799, 765, 737, 715, 672, 646, 517, 471.

**General method for the synthesis of [Mg(Tptm)(X)] (1-X; X = Cl, Br, I):** Under an atmosphere of dry argon, tris(2-mercaptopyridyl)methane (HTptm, 0.686 g, 2.0 mmol) was weighed into a flamed Schlenk flask, and diethyl ether was added (40 mL) with stirring to dissolve the reagent. A solution in Et_2_O or THF of the corresponding Grignard reagent (2.0 mmol) was then added, and after stirring for 1 h at room temperature, the solution was filtered. The precipitate was dried to yield pale yellow powder. This was recrystallized from boiling toluene (X = Br, I) or benzene (X = Cl).

**1-Cl:** From *^i^*PrMgCl (1.0 mL, 2.0 mmol, 2.0 M in THF). Yield: 0.787 g (98%). ^1^H NMR (400 MHz, 298 K, C_6_D_6_): δ_H_ (ppm) = 6.19 (3H, m, C_5_H_3_N-C*H*^b^), 6.47 (6H, m, C_5_H_3_N-C*H*^c^ and C_5_H_3_N-C*H*^d^), 9.61 (3H, d, ^3^*J*_HH_ = 5.6 Hz, C_5_H_3_N-C*H*^a^). ^13^C{^1^H} NMR (125 MHz, 298 K, C_4_D_8_O): δ_C_ (ppm) = 15.0 (Mg*C*), 118.7 (C_5_H_3_N-*C*^b^H), 120.0 (C_5_H_3_N-*C*^c^H), 137.9 (C_5_H_3_N-*C*^d^H), 150.7 (C_5_H_3_N-*C*^a^H), 165.3 (C_5_H_3_N-*C*^e^H). Anal. calcd. for C_32_H_24_Cl_2_Mg_2_N_6_S_6_∙C_4_H_8_O: C, 49.27%; H, 3.79%; N, 9.58%. Found: C, 49.74%; H, 3.99%; N, 9.20%. FTIR: $\tilde{\nu }$ (cm^−1^) = 2918, 2852, 1585, 1554, 1456, 1413, 1283, 1131, 1045, 1003, 755, 720, 638, 586, 486, 412.

**1-Br:** From MeMgBr (0.238 g, 2.00 mmol, dissolved in Et_2_O). Yield: 0.805 g (90%). ^1^H NMR (400 MHz, 298 K, C_6_D_6_): δ_H_ (ppm) = 6.41 (3H, t, ^3^*J*_HH_ = 6.4 Hz, C_5_H_3_N-C*H*^b^), 7.25 (3H, d, ^3^*J*_HH_ = 7.9 Hz, C_5_H_3_N-C*H*^d^), 7.35, (3H, d, ^3^*J*_HH_ = 8.0 Hz, C_5_H_3_N-C*H*^c^), 8.22 (3H, d, ^3^*J*_HH_ = 4.3 Hz, C_5_H_3_N-C*H*^a^). ^13^C{^1^H} NMR (125 MHz, 298 K, C_4_D_8_O): δ_C_ (ppm) = 15.1 (Mg*C*), 118.7 (C_5_H_3_N-*C*^b^H), 120.2 (C_5_H_3_N-*C*^c^H), 138.1 (C_5_H_3_N-*C*^d^H), 151.3 (C_5_H_3_N-*C*^a^H), 180.7 (C_5_H_3_N-*C*^e^H). Anal. calcd. for C_32_H_24_Br_2_Mg_2_N_6_S_6_∙C_4_H_10_O: C, 44.69%; H, 3.54%; N, 8.69%. Found: C, 44.48%; H, 3.27%; N, 8.46%. FTIR: $\tilde{\nu }$ (cm^−1^) = 3062, 2969, 2582, 1587, 1555, 1456, 1415, 1283, 1155, 1131, 1091, 1046, 1003, 766, 756, 721, 638, 592, 485, 411.

**1-I:** From MeMgI (0.332 g, 2.00 mmol, dissolved in Et_2_O). Yield: 938 mg (95%). ^1^H NMR (400 MHz, 298 K, C_6_D_6_): δ_H_ (ppm) = 6.16 (3H, m, C_5_H_3_N-C*H*^b^), 6.42 (6H, m, C_5_H_3_N-C*H*^c^ and C_5_H_3_N-C*H*^d^), 9.88 (3H, d, ^3^*J*_HH_ = 5.7 Hz, C_5_H_3_N-C*H*^a^). ^13^C{^1^H} NMR (100 MHz, 298 K, C_6_D_6_): δ_C_ (ppm) = 33.0 (Mg*C*), 118.6 (C_5_H_3_N-*C*^b^H), 120.4 (C_5_H_3_N-*C*^c^H), 138.0 (C_5_H_3_N-*C*^d^H), 151.8 (C_5_H_3_N-*C*^a^H), 164.8 (C_5_H_3_N-*C*^e^H). Anal. calcd. for C_32_H_25_I**_2_**Mg**_2_**N_6_S_6_∙0.5(C_4_H_10_O): C, 39.82%; H, 2.95%; N, 8.20%. Found: C, 39.50%; H, 2.65%; N, 8.04%. FTIR: $\tilde{\nu }$ (cm^−1^) = 3066, 3009, 2972, 2865, 1588, 1555, 1458, 1413, 1283, 1131, 1091, 1046, 1002, 960, 879, 768, 756, 723, 640, 597, 484, 410.

**2:** To a flamed Schlenk flask was added 1-I (0.493 g, 1.00 mmol), benzyl potassium (0.130 g, 1.00 mmol), and tetrahydrofuran (40 mL). The reaction mixture was stirred overnight at room temperature, yielding a red solution. The precipitate was filtered off, and the solvent was removed from the filtrate in vacuo. The residue was extracted with benzene, and the solvent was removed in vacuo. The crude product was recrystallized from toluene, affording a small crop of crystals of 2 (ca. 20 mg). ^1^H NMR (400 MHz, 298 K, C_6_D_6_): δ_H_ (ppm) = 5.15 (2H, s, MgC*H*), 5.98 (4H, t, ^3^*J*_HH_ = 6.3 Hz, C_5_H_3_N-C*H*^b^), 6.18 (4H, t, ^3^*J*_HH_ = 7.7 Hz, HC(C_5_H_3_N-C*H*^d^), 6.52 (4H, d, ^3^*J*_HH_ = 8.2 Hz, C_5_H_3_N-C*H*^c^), 8.32 (4H, d, ^3^*J*_HH_ = 5.1 Hz, C_5_H_3_N-C*H*^a^). ^13^C{^1^H} NMR (100 MHz, 298 K, C_6_D_6_): δ_C_ (ppm) = 20.0 (Mg*C*H), 117.9 (C_5_H_3_N-*C*^b^H), 121.6 (C_5_H_3_N-*C*^d^H), 135.9 (C_5_H_3_N-*C*^c^H), 147.2 (C_5_H_3_N-*C*^a^H), 166.5 (C_5_H_3_N-*C*^e^H).

**(C_5_H_4_N-S)_2_C=C(C_5_H_4_N-S)_2_:** Tris(2-mercaptopyridyl)methane (HTptm, 686 mg, 2.0 mmol) and potassium bis(trimethylsilyl)amide (378 mg, 2.0 mmol) were weighed into a flamed Schlenk flask, and toluene added (50 mL) with stirring to dissolve the reagents. After stirring for 1 h, a chestnut precipitate formed, and the reaction mixture was transferred to the freezer overnight. The precipitate was filtered off, and recrystallization from toluene afforded a small crop of crystals of (C_5_H_4_N-S)_2_C=C(C_5_H_4_N-S)_2_ (ca. 10 mg), sufficient only for XRD characterisation.

### 6.3. Computational Methods

DFT geometry optimizations were run with Gaussian 16 (Revision A.03) [34] using the BP86 functional [35,36]. Si, S and K centres were described with the Stuttgart RECPs and associated basis sets [37], and 6–31G** basis sets were used for all other atoms [38,39]. A set of *d*-orbital polarization functions was also added to Si (ζ^d^ = 0.284), S (ζ^d^ = 0.503) and K (ζ^d^ = 1.000) [40]. Stationary points were characterized by analytical frequency calculations that also provided the thermochemical corrections for the final free energies reported in the text. Electronic energies were re-computed using the triple-ζ basis set Def2-TZVP [41,42] and included corrections for dispersion using the D3BJ method [43] and solvation in toluene using PCM [44]. All geometries are supplied as a separate XYZ file.

### 6.4. Crystallographic Methods

The crystal data for all compounds are compiled in Appendix A. Crystals of **1-Br****, 1-I, 1-I·THF, 2** and (C_5_H_4_N-S)_2_C=C(C_5_H_4_N-S)_2_ were examined using a Bruker Apex 2000 CCD area detector diffractometer, and data were collected using graphite-monochromated Mo-K*α* radiation (*λ* = 0.71073). Crystals of **1-Cl** were examined using a dual-wavelength [Mo-K*α* (*λ* = 0.71073) or Cu-K*α* (*λ* = 1.54178)] Rigaku FR-X diffractometer with a HyPix 6000HE photon-counting detector. Intensities were integrated from data recorded on 1°frames by *ω* rotation. A multiscan method (SADABS) [45] or a Gaussian grid faced-indexed absorption correction with a beam profile were applied [46]. The structures were solved using SHELXS [47]; the datasets were refined by full-matrix least-squares on reflections with *F^2^ ≥ 2σ(F^2^)* values, with anisotropic displacement parameters for all non-hydrogen atoms, and with constrained riding hydrogen geometries [48]. Uiso(H) was set at 1.2 (1.5 for methyl groups) times Ueq of the parent atom. The largest features in the final difference syntheses were close to heavy atoms and were of no chemical significance. SHELX [47,48] was employed through OLEX2 for structure solution and refinement [49]. ORTEP-3 [50] and POV-Ray [51] were employed for molecular graphics. The structures have been deposited within the Cambridge Crystallographic Data Centre (CCDC 2088557, 2175279–2175282, 2175291, 2175339). This information can be obtained free of charge from www.ccdc.cam.ac.uk/data_request/cif (accessed on 13 July 2022). Inspection of the data of **1-I·THF** revealed the presence of a non-merohedral twin component accounting for approximately 40% of the reflections. The partially refined data were treated with PLATON TWINROTMAT [52] to separate the two twin components and generate a file containing reflection information from both of them. The structure was then refined against this new reflection file, and the relative scale factor of the two components was refined, converging at a ratio of 0.432(4):0.568(4).

## Data Availability

Additional research data supporting this publication are available from Mendeley at doi: 10.17632/694vs8nmgp.3.

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
