# Peer review of "Structural Investigation of Magnesium Complexes Supported by a Thiopyridyl Scorpionate Ligand"

_molecules, 2022, doi:10.3390/molecules27144564_

Round 1

Reviewer 1 Report

The manuscript by Stevens et al describes the synthesis and characterization of novel thiopyridyl ligands and their complexes with magnesium. All new compounds are characterized by spectroscopic and other analytical methods, including X-ray crystallography. The manuscript is accurate, well-structured and provides exhaustive supporting data (datapoints and raw data for IR and NMR spectra etc). The work contributes to the modern coordination and organometallic chemistry and can be recommended for publication after correction of minor issues described below.

1) The CheckCIF 1-I_THF.pdf demonstrates the presence of twinning, as described in the Authors’ response to Alert level B. However, no twinning refinement is mentioned in the main text or experimental section. Also, despite the files 1-Br.cif and 1-I-THF.cif show the same alert – “Low Bond Precision on C-C Bonds”, the response is given only for the latter CIF. Please add responses to all alerts of the B level.

2) Lines 199, 202, 205, 206, 217, 221, 238, 249, 264 (Figure 3 caption): the phrase “Error! Reference source not found” looks like the software problem. Please check and correct.

3) Figures 3 and S19 seem to be slightly distorted horizontally (elongated), please check if it is the case.

4) The phrase “Electronic energies ... include corrections for dispersion using the D3BJ method” can be interpreted as the absence of dispersion correction for other calculations, e.g. geometry optimization. If so, correction for dispersion should be introduced as it may have notable influence on the geometry of aromatic groups.

Author Response

We thank the reviewer for taking their time to review our work and for their useful feedback. This document includes our response to the reviewer's queries (in bold).

The manuscript by Stevens et al describes the synthesis and characterization of novel thiopyridyl ligands and their complexes with magnesium. All new compounds are characterized by spectroscopic and other analytical methods, including X-ray crystallography. The manuscript is accurate, well-structured and provides exhaustive supporting data (datapoints and raw data for IR and NMR spectra etc). The work contributes to the modern coordination and organometallic chemistry and can be recommended for publication after correction of minor issues described below.

1) The CheckCIF 1-I_THF.pdf demonstrates the presence of twinning, as described in the Authors’ response to Alert level B. However, no twinning refinement is mentioned in the main text or experimental section. Also, despite the files 1-Br.cif and 1-I-THF.cif show the same alert – “Low Bond Precision on C-C Bonds”, the response is given only for the latter CIF. Please add responses to all alerts of the B level.

Information about the twinning treatment were originally incorporated in the refine_special_details section of the .cif file. For clarity, we have now included this information in the ‘Crystallographic method’ section. With regards to the B level alerts in 1-Br, we wrongly uploaded an old cif check report. All B level alerts have been addressed and updated files have been provided.

2) Lines 199, 202, 205, 206, 217, 221, 238, 249, 264 (Figure 3 caption): the phrase “Error! Reference source not found” looks like the software problem. Please check and correct.

All these errors have been checked.

3) Figures 3 and S19 seem to be slightly distorted horizontally (elongated), please check if it is the case.

This has been addressed and new figures have been produced.

4) The phrase “Electronic energies ... include corrections for dispersion using the D3BJ method” can be interpreted as the absence of dispersion correction for other calculations, e.g. geometry optimization. If so, correction for dispersion should be introduced as it may have notable influence on the geometry of aromatic groups.

As pointed out by the reviewer, the geometry optimizations were performed in the absence of dispersion. Following the reviewer’s comments, these were re-optimized using D3BJ. Structures containing toluene rings coordinated to the K atom converged into unrealistic geometrics due to overestimation of ligand-ligand interactions. e.g. the measured N-K-C7H8 angle in [K{N(SiMe3)2}(η6-C7H8)] in the absence of dispersion was 163.0 ° (measured using the centroid of the toluene ring). This is comparable to angles found in related species for which structural data is available such as [K(η6-C7H8)2]+ , where the C7H8-K-C7H8 angle is 156.0 ° (Inorg. Chem. 2017, 56, 5959). However, the optimisation of [K(η6-C7H8)2]+  with G3BJ dispersion induces a substantial bending of the aforementioned angle into 107.9 °. The N-K-C7H8 angle in [K{N(SiMe3)2}(η6-C7H8)] is bent even more drastically; from is 163.0 ° to only 90.4 °. We therefore believe that the application of post-SCF dispersion corrections, as we initially reported, is more correct for these particular systems.

Reviewer 2 Report

The paper by Sotorios and Ortu et. al represents research focusing on the synthesis of scorpionate ligand and magnesium salts based on them. Such types of ligands are well known and their synthesis is routine work. But new Mg complexes and investigation of volume buried is novel. The investigation seems to be suitable for publication in Molecules but needs major revisions:

- Another group published the first synthesis of this type of scorpionate ligand (see 10.1016/S0022-2860(01)00976-0.) This fact has been discussed by Kinoshita et al. Using ultrasonic irradiation could help to arise the yields.

- The principal axes used for volume buried steric maps on the example of one structure are suggested being given. Correspondingly these axes could be presented on the steric maps for a better understanding.

-Check the titles of the figures. The phrase “Error! Reference source not found” seems to be incorrect

- The NMR studies should be carefully checked because in all cases the benzene-d6 is presented as a solvent, but only in the several cases, it is presented on spectra. For example figure s5, there are the signals of C6D6 in the 1H NMR spectrum, but it is not presented in the 13 C spectrum. The presence of a significant amount of solvents (THF and Et2O) raises a question about the quality of elemental analysis.

- The IR spectra presented in SI (Fig S17-18) also seem to be incorrect. In the case of using an ATR module, the transmittance spectra are measured, not absorbance. There are the problems with baseline, please correct or explain.

Author Response

We thank the reviewer for taking their time to review our work and for their feedback. Here we provide a point-by-point response (in bold) to the reviewer's queries.

The paper by Sotorios and Ortu et. al represents research focusing on the synthesis of scorpionate ligand and magnesium salts based on them. Such types of ligands are well known and their synthesis is routine work. But new Mg complexes and investigation of volume buried is novel. The investigation seems to be suitable for publication in Molecules but needs major revisions:

- Another group published the first synthesis of this type of scorpionate ligand (see 10.1016/S0022-2860(01)00976-0.) This fact has been discussed by Kinoshita et al. Using ultrasonic irradiation could help to arise the yields.

We thank the reviewer for the suggestion; this was actually an oversight on our behalf as we had indeed tried to use the method illustrated by de Castro et al. and we now added the reference and a sentence to highlight this. We also thank the reviewer for the suggestion regarding the methodology employed by Kinoshita et al.; in fact, this is something we had considered but decided not to implement at this time because of the reliability of Parkin’s methodology. We have nowadded a sentence in the manuscript to highlight this.

- The principal axes used for volume buried steric maps on the example of one structure are suggested being given. Correspondingly these axes could be presented on the steric maps for a better understanding.

We have added a new figure (Figure 7) which shows the principal axis used for the buried volume calculations and the viewpoint for the steric maps.

-Check the titles of the figures. The phrase “Error! Reference source not found” seems to be incorrect

All figure titles have been checked.

- The NMR studies should be carefully checked because in all cases the benzene-d6 is presented as a solvent, but only in the several cases, it is presented on spectra. For example figure s5, there are the signals of C6D6 in the 1H NMR spectrum, but it is not presented in the 13 C spectrum. The presence of a significant amount of solvents (THF and Et2O) raises a question about the quality of elemental analysis.

We apologise for the oversight in the presentation of the NMR data. This issue has now been resolved and NMR spectra have been assigned correctly. We had to use d8-THF for acquiring 13C spectra due to the poor solubility of some of the compound in d6-benzene – we required very concentrated samples to identify the signal of the methanide carbon. Both versions of the spectra (13C in d6-benzene and d8-THF) are now reported for complete clarity.

With regards to the elemental analysis results, there are only small amounts of solvent present (one or half a molecule of THF/Et2O per dimer) and these are the solvents employed in the preparation of these samples. These analyses are also corroborated by the 1H NMR analysis, which are in very good agreement with the bulk purity of our samples and show the presence of the same solvents detected by the EA analyses.

- The IR spectra presented in SI (Fig S17-18) also seem to be incorrect. In the case of using an ATR module, the transmittance spectra are measured, not absorbance. There are the problems with baseline, please correct or explain.

We thank the reviewer for spotting this: there was indeed an issue with the data. We have now recollected these spectra and the new data is reported in the SI.

Round 2

Reviewer 2 Report

The paper could be published